# Cytosolic thioredoxin reductase 1 is required for correct disulfide formation in the ER

Greg J Poet[1],[†] (ID), Ojore BV Oka[1], Marcel van Lith[1], Zhenbo Cao[1], Philip J Robinson[1], Marie Anne Pringle[1], Elias SJ Arnér[2] (ID) & Neil J Bulleid[1],[*] (ID)

## Abstract

Folding of proteins entering the secretory pathway in mammalian cells frequently requires the insertion of disulfide bonds. Disulfide insertion can result in covalent linkages found in the native structure as well as those that are not, so-called non-native disulfides. The pathways for disulfide formation are well characterized, but our understanding of how non-native disulfides are reduced so that the correct or native disulfides can form is poor. Here, we use a novel assay to demonstrate that the reduction in non-native disulfides requires NADPH as the ultimate electron donor, and a robust cytosolic thioredoxin system, driven by thioredoxin reductase 1 (TrxR1 or TXNRD1). Inhibition of this reductive pathway prevents the correct folding and secretion of proteins that are known to form non-native disulfides during their folding. Hence, we have shown for the first time that mammalian cells have a pathway for transferring reducing equivalents from the cytosol to the ER, which is required to ensure correct disulfide formation in proteins entering the secretory pathway.

**Keywords** disulfide formation; endoplasmic reticulum; protein disulfide isomerase; redox homeostasis; thioredoxin reductase
**Subject Categories** Membrane & Intracellular Transport; Protein Biosynthesis & Quality Control
The EMBO Journal (2017) 36: 693–702

## Introduction

The formation of disulfides within the mammalian endoplasmic reticulum (ER) is a highly orchestrated process that occurs during the folding and assembly of proteins entering the secretory pathway (Braakman & Bulleid, 2011). The introduction of disulfides is catalyzed by a family of disulfide exchange proteins typified by protein disulfide isomerase (PDI; Ellgaard & Ruddock, 2005). The disulfide within their active sites is exchanged with free thiols within a folding polypeptide chain as it enters the lumen of the ER. The

mechanism for the reoxidation of the PDI active site can involve multiple pathways that couple the introduction of a disulfide with the reduction in an electron acceptor such as oxygen or hydrogen peroxide (Bulleid & Ellgaard, 2011). The disulfide formed within the folding polypeptide may be the same as that found in the native state, but can also form between two thiols not linked in the final structure (Jansens et al, 2002). Such non-native disulfides need to be resolved for the polypeptide to attain its native state. In addition, disulfides in proteins that are targeted for degradation may be reduced prior to their dislocation from the ER to the cytosol (Ushioda et al, 2008; He et al, 2015). Hence, in parallel to an oxidative pathway, a reductive pathway must exist in the ER to ensure correct protein folding and allow dislocation of proteins for degradation.

The initial stage in the reduction in non-native disulfides is catalyzed by members of the PDI family such as ERdj5 (Ushioda et al, 2008; Oka et al, 2013). This enzyme is characterized by four thioredoxin (Trx) fold domains which have active site dithiol/disulfides motifs with relatively low reduction potential making it an efficient reductase and the presence of a J domain which can bind to the ER chaperone BiP (Cunnea et al, 2003; Hagiwara et al, 2011). Substrates are likely to be targeted to ERdj5 due to their binding to BiP via hydrophobic regions which may be exposed during folding (Fourie et al, 1994). Many of the substrates for ERdj5 contain several independent folding domains as well as multiple disulfides increasing their potential to interact with BiP and form non-native disulfides. Most notably, formation of the active site disulfides and a functional J domain within ERdj5 are required for the correct folding of substrate proteins (Oka et al, 2013).

During catalysis, the ERdj5 active site dithiol motifs will be oxidized to disulfides and, therefore, require subsequent reduction to allow further enzymatic activity. Within the cytosol, the main disulfide reducing protein thioredoxin 1 (Trx1) is maintained in a reduced state by the action of cytosolic TrxR1 with potential contribution from both the glutathione reductase (GR) and glutaredoxin (Grx) pathways (Lillig & Holmgren, 2007). However, there are no known equivalent pathways present within the ER for the reduction in disulfides in the Trx domains within the members of the PDI family. A potential GR variant does exist in the ER, called ERFAD (ER flavoprotein associated with degradation); however, it does not

1  The Institute of Molecular, Cell and Systems Biology, CMVLS, University of Glasgow, Glasgow, UK
2  Division of Biochemistry, Department of Medical Biochemistry and Biophysics (MBB), Karolinska Institutet, Stockholm, Sweden
   *Corresponding author. Tel: +44 141 330 3870; Fax: +44 141 330 5481; E-mail: neil.bulleid@glasgow.ac.uk
   †Present address: St. Jude Children's Research Hospital, Memphis, TN, USA

contain the active site cysteines essential for function as GR or TrxR (Riemer *et al*, 2009). In prokaryotes, disulfide formation and reduction within the periplasmic space is catalyzed by the disulfide exchange proteins DsbA and DsbC, which are structurally homologous to Trx1 (Kadokura *et al*, 2003). Disulfide formation is coupled to the electron transport chain via a membrane protein called DsbB, allowing *de novo* disulfide formation in the disulfide exchange protein DsbA which catalyzes disulfide formation in substrate proteins. The reductive pathway is coupled to bacterial TrxR within the cytosol, by DsbD transferring electrons across the membrane to reduce the disulfide exchange protein DsbC that finally catalyzes disulfide reduction.

Previous attempts to identify the ER reductive pathway have focused on the role of glutathione (GSH) in the process (Chakravarthi *et al*, 2006). In particular, GSH has been shown to play an important role in buffering the capacity of the ER from fluctuations in reductive or oxidative stress ensuring a rapid re-equilibration of the redox status of several members of the PDI family (Jessop & Bulleid, 2004; Molteni *et al*, 2004; Appenzeller-Herzog *et al*, 2010). However, its requirement for the formation of the correct disulfides in proteins is less clear. Depletion of ER GSH either by inhibition of GSH synthase or by targeting GSH-degrading enzymes to this organelle did not prevent correct disulfide formation in proteins containing complex disulfides such as tissue-type plasminogen activator or the LDL-receptor (Chakravarthi & Bulleid, 2004; Tsunoda *et al*, 2014). Even if GSH is involved in the reductive pathway by reducing ERdj5, the resulting glutathione disulfide (GSSG) formed would still need to be reduced either enzymatically in the ER or following transport back into the cytosol, by reaction with free thiols in proteins entering the ER or by oxidation of other PDI family members (Bulleid & Ellgaard, 2011).

To provide insight into the reductive pathway involved in ensuring that correct disulfides are formed within proteins, we took advantage of a cell-free translation system that allowed us to selectively manipulate the cytosolic reductive pathway. Importantly, we have developed a system that allows us to assay the transition from non-native to native disulfides during protein synthesis. Using this approach, we were able to determine the consequence of manipulation of the individual cytosolic reductive pathways on formation of the correct disulfides in proteins entering the mammalian ER.

# Results

### The role of the Trx pathway in preventing disulfide formation in the cytosol

Previous characterization of rabbit reticulocyte lysate systems for cell-free translation has highlighted the requirement for reducing equivalents to ensure efficient and sustained protein synthesis (Jackson *et al*, 1983a,b). As a consequence of these studies, most commercial lysates contain added reducing agents such as dithiothreitol (DTT), which maintains protein synthesis but also prevents disulfide formation. To be able to evaluate the endogenous reductive pathway during cell-free translation, we made use of a commercial lysate that does not contain added DTT but still shows high levels of protein synthesis (Wilson *et al*, 1995). When a variety of protein transcripts were translated and the products separated by SDS–PAGE under non-reducing conditions, it was noted that all

formed intra- and to some extent inter-chain disulfides as judged by mobility shifts compared to the same samples run under reducing conditions (Fig 1A–D, compare lanes 1 and 2). These transcripts code for proteins normally located in the cytosol (c-VIMP, a soluble Sec-to-Cys variant of SelS/VIMP lacking the transmembrane domain, Fig 1A; Christensen *et al*, 2012) or the ER (influenza virus hemagglutinin (HA), prolactin and β1-integrin, Fig 1B–D). This result was unexpected as protein disulfides do not normally form within the cytosol of mammalian cells and indicated that the reducing capacity of the reticulocyte cytosol extract had become compromised during its preparation.

There are two major pathways that might prevent disulfide formation in proteins synthesized in the cytosol, namely the Trx1/TrxR1 or the GSH/GR/Grx pathway (Fig 2; Lu & Holmgren, 2014). Both of these pathways use NADPH as the ultimate electron donor so aberrant disulfide formation in the reticulocyte lysate could be due to a lack of NADPH. The main mechanism for NADPH recycling in the cytosol is through the action of glucose-6-phosphate dehydrogenase (G6PDH), so to determine whether replenishing NADPH could prevent disulfide formation, we included increasing concentration of G6P during translation and evaluated the redox status of the synthesized proteins (Fig 1). In all cases, disulfide formation was prevented when the added G6P concentration exceeded 0.2 mM. This result demonstrates that NADPH is the most likely electron donor and that we can re-establish the reducing environment in the cytosolic extract by the addition of G6P.

To determine which of the reductive pathways was involved in preventing disulfide formation, we made use of inhibitors of G6PDH (dehydroepiandrosterone—DHEA), TrxR1 (auranofin), or GR (carmustine). For these experiments, we used β1-integrin as it contains several thiols capable of forming disulfides (Fig 3). Similar results were observed when HA was translated in the presence of these inhibitors (Fig EV1). Inhibiting G6PDH prevented the restorative effect of G6P demonstrating that G6P prevents disulfide formation by the regeneration of NADPH during its conversion to 6-phoshogluconolactone (Fig 3A). As DHEA is a competitive inhibitor, its effects are noticeable but not complete. A more dramatic effect was seen when TrxR1 was irreversibly inhibited with auranofin (Fig 3B) at a concentration that prevents the activity of the purified enzyme (Fig EV2). Even at low μM concentrations, well below the concentrations required to inhibit GR (Gromer *et al*, 1998; Fig EV2), the restorative effect of G6P was reversed allowing disulfides to be formed (lanes 3–6). Remarkably, the GR inhibitor carmustine had no effect on the disulfide reduction mediated by G6P at concentrations known to inhibit the enzyme (Figs 3C and EV2; Liu & Sturla, 2009). Some slight inhibition occurred at 3.66 mM with HA (Fig EV1) though it should be noted that carmustine also inhibits TrxR1 at higher concentrations (Fig EV2; Witte *et al*, 2005). Together these results suggest that the main pathway that prevents disulfides forming in proteins synthesized within a reticulocyte lysate is mediated by TrxR1 and not GR.

To evaluate further the role of the individual cytosolic reductive pathways in preventing disulfide formation, we determined the changes that occurred to either the Trx1 redox status or the GSSG/GSH ratio after addition of G6P to the lysate. To determine the Trx1 redox state, we took advantage of the fact that modification of free thiols with the alkylating agent 4-acetamido-4′-maleimidylstilbene-2,2′-disulfonic acid (AMS) results in a decrease in electrophoretic mobility of modified relative to unmodified proteins. We added

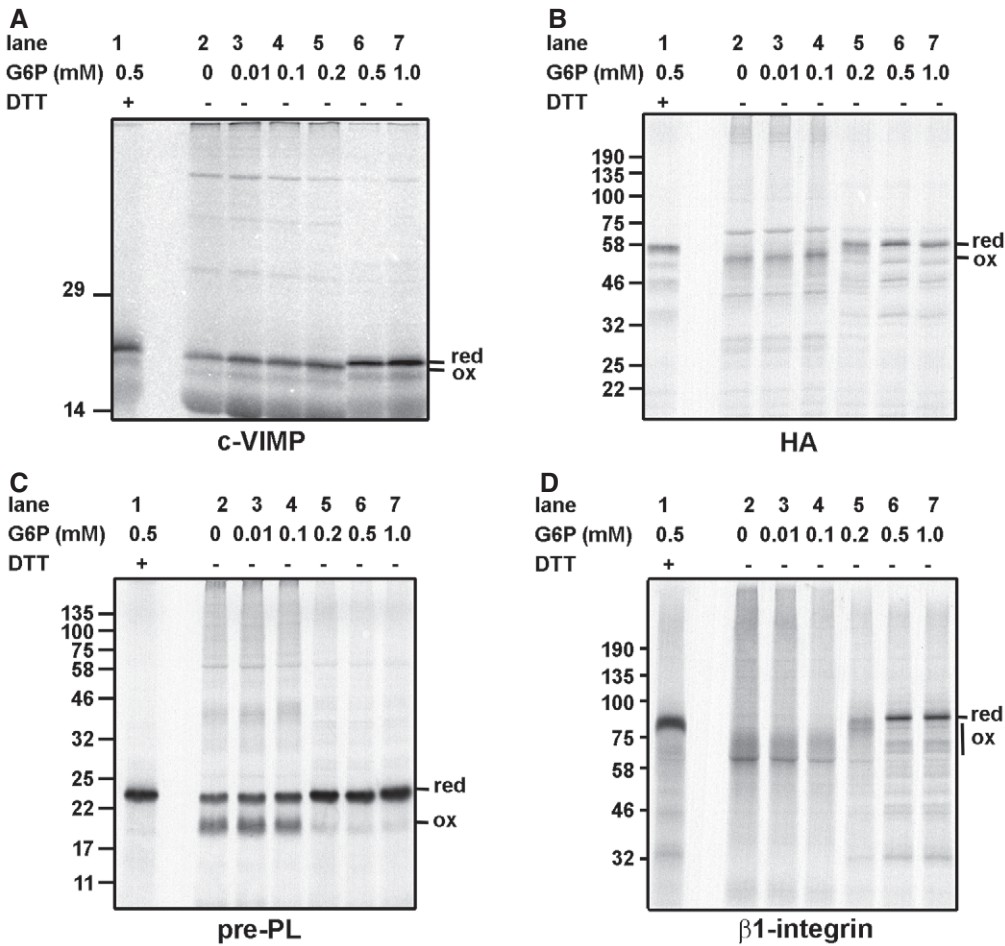

**Figure 1. The formation of disulfides during cell-free translation of several proteins can be prevented by the addition of G6P.**

A–D Cell-free translation with $^{35}$S-labeling of newly synthesized protein was carried out in the presence of increasing concentrations of G6P as indicated. The transcripts used were (A) c-VIMP, (B) influenza virus HA, (C) preprolactin, and (D) β1-integrin. The samples were separated by SDS–PAGE under reducing (lane 1) or non-reducing (lanes 2–7) conditions. The mobility of the reduced (red) or oxidized (ox) protein is as indicated.

purified streptavidin binding peptide-tagged Trx1 to the lysate as the migration of the endogenous protein by SDS–PAGE was distorted by the high concentration of globin in the lysate. We treated lysates, either before or after incubation with G6P, with NEM to block free thiols, and subsequently reduced and then alkylated with AMS so that slower migrating bands indicate oxidized protein. In the untreated sample, two bands were revealed for Trx indicating that in the absence of G6P, Trx1 was partially oxidized (Fig 4A, lane 1). After the lysate was treated with G6P, only a single Trx1 species was present which migrated with the reduced form, indicating that a consequence of the addition of G6P was the reduction in Trx1 (lane 2). We determined the redox state of Trx1 after G6P addition but in the presence of auranofin (lane 3). Trx1 remained as a mixture of oxidized and reduced forms demonstrating that the reduction in Trx1 mediated by G6P addition required TrxR1.

To determine whether the consequence of G6P addition was to alter the GSSG/GSH ratio, thereby preventing disulfide formation, we monitored the change in GSSG/GSH ratio following G6P addition (Fig EV2). Even in the absence of G6P, the ratio of GSSG/GSH was low indicating that the lysate contained predominantly GSH. The ratio did not change markedly following G6P addition indicating that the ability of the lysate to support disulfide formation was not a consequence of changes in the GSH redox balance.

Finally, to further determine the role of GSH in modulating disulfide formation, we depleted GSH from the lysate. To remove GSH, we added a GSH-degrading enzyme called ChaC1 to the reticulocyte lysate (Kumar *et al*, 2012). The total GSH levels were depleted to < 10% following treatment (Fig 4B). It is likely that the residual amount of GSH remaining is the oxidized form as ChaC1 does not degrade GSSG (Tsunoda *et al*, 2014). The reticulocyte lysate was still active in protein synthesis with extensive disulfide formation occurring in the absence of added G6P using β1-integrin as substrate (Fig 4C, lane 4). However, the addition of G6P prevented most of this disulfide formation (lane 5). Hence, the removal of GSH does not prevent the G6P-mediated reduction in disulfides.

Taken together, these results show a demonstrable effect of G6P addition on the redox status of Trx1 and suggest that the pathway for reduction in disulfides in the cytosol is maintained by the recycling of NADP to NADPH, which drives the reduction in Trx1 by TrxR1. The lack of change to the GSSG/GSH ratio and the

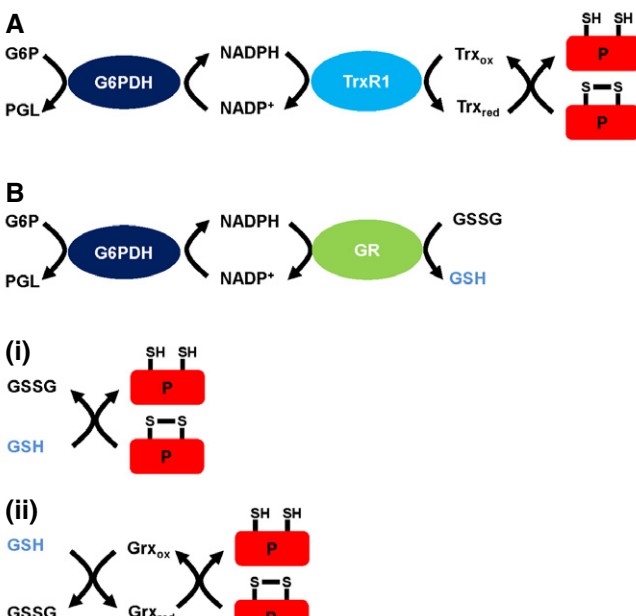

**Figure 2. Pathways for disulfide reduction in the cytosol.**

A The TrxR1 pathway is driven by NADPH reducing Trx. The reduced Trx can then efficiently reduce disulfides in substrate proteins (P).

B The GR pathway is also driven by NADPH resulting in the reduction in glutathione disulfide. The GSH generated can either (i) directly reduce disulfides in substrate proteins or (ii) reduce glutaredoxin (Grx) which then reduces substrate proteins.

continuation of the reductive pathway following GSH depletion would suggest that GR/GSH/Grx pathway is not responsible for preventing disulfide formation in the cytosol, at least under the conditions studied here. It should be noted, however, that it has

been shown previously that there is potential cross talk between the Grx and Trx1 pathways (Reichheld *et al*, 2007).

## The cytosolic reductive pathway drives the resolution of non-native disulfides formed within the ER

We describe above the development of a cell-free translation system that prevents disulfides forming in newly synthesized proteins without the addition of exogenous reducing agents. To determine the ability of the same system to support disulfide formation in proteins translocated into the ER, we supplemented the lysate with semi-permeabilized (SP) cells (Wilson *et al*, 1995). As previously described (Jessop *et al*, 2007), when β1-integrin was translated in the presence of SP cells an additional, slower migrating product was observed that corresponds to the glycosylated protein (Fig 5A, lane 1). When the same translation products were separated under non-reducing conditions, a different migration pattern was observed depending on the presence or absence of added G6P (Fig 5A, lanes 5 and 6). In the absence of G6P, a broad smear migrating faster than the reduced glycosylated protein as well as higher molecular weight aggregates was observed (lane 5). The smear would indicate that disulfides had formed, but a heterogeneous mixture of species was present containing correct and incorrect disulfides. As β1-integrin contains 21 disulfides, it has a high potential to form the incorrect disulfides. However, in the presence of G6P, a more distinct, faster migrating product was observed with little evidence of high molecular weight aggregates (lane 6). Importantly, disulfides were formed under conditions where no disulfides were formed in the absence of SP cells (Fig 1D, lane 7). Indeed, the untranslocated protein seen when translations were carried out in SP cells was also fully reduced in the presence of G6P (Fig 5A, lane 6). Note that the mobility of the untranslocated protein separated under reducing conditions is different after translation in the absence or presence of G6P (Fig 5A, lanes 1 and 2). This is most likely due to modification of free thiols by NEM slowing the

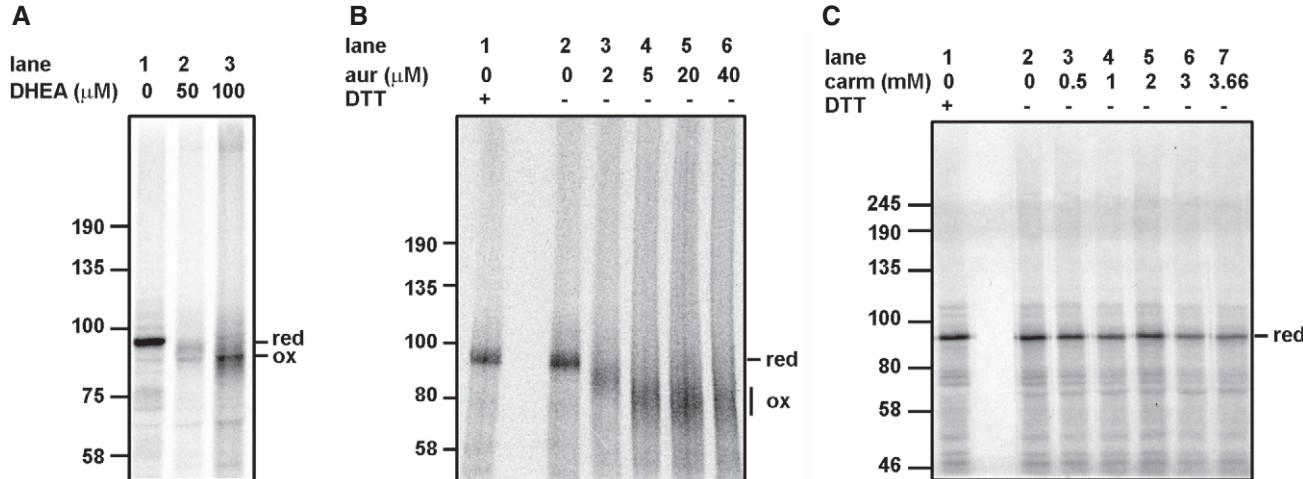

**Figure 3. Inhibition of G6PDH or TrxR1 but not GR allows disulfide formation in the presence of G6P.**

A–C Cell-free translation of β1-integrin was carried out in the presence of G6P and increasing concentrations of (A) DHEA, (B) auranofin (aur), or (C) carmustine (carm) as indicated. The translation mixture was incubated for 10 min prior to the reaction being initiated by the addition of mRNA and incubation at 30°C. Translation products were separated by SDS–PAGE after prior reduction with DTT (B and C, lane 1) or without reduction (A, lanes 1–3 or B and C, lanes 2–6).

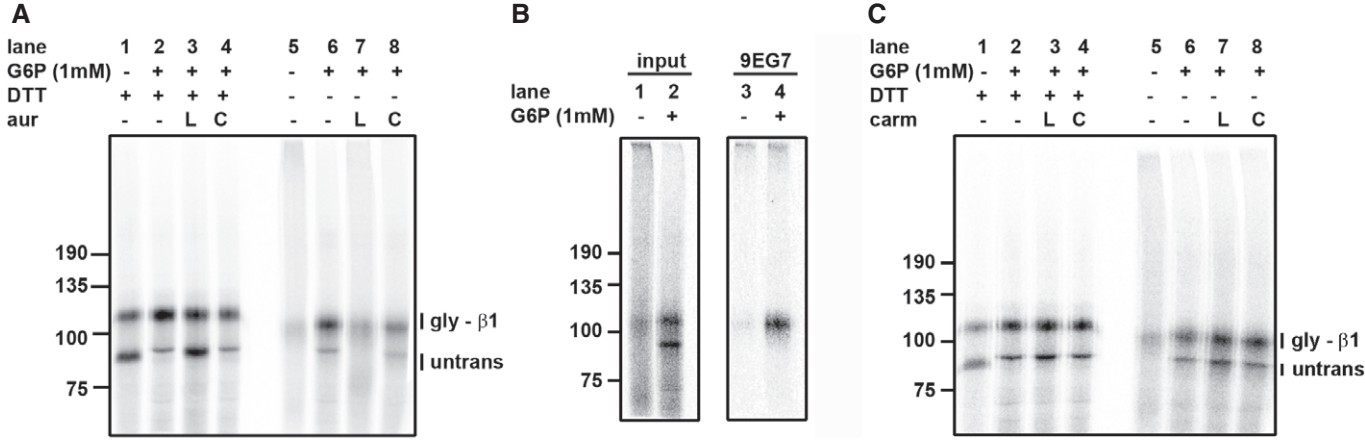

**Figure 4. Trx1 is reduced following G6P addition, and GSH depletion does not prevent the G6P-mediated reduction in disulfides.**

A The oxidation status of added Trx1 was assayed through immunoblotting following a 60-min incubation in a cell-free translation system in the absence or presence of G6P with and without auranofin (aur) as indicated. Following incubation, Trx1 was isolated, reduced, and any free thiols released were modified with AMS which retards electrophoretic mobility. Hence, the slower migrating band of the doublet is Trx1 that was oxidized in the original sample. The position of the reduced (red) or oxidized (ox) protein is as indicated.

B Reticulocyte lysate was treated with or without 5 μM purified ChaC1 for 60 min prior to measuring the GSH concentration. The results are the average from three separate determinations ± SD.

C Cell-free translation of β1 integrin was carried out in the presence or absence of G6P as indicated and in normal (lanes 1–3) or GSH-depleted (lanes 4 and 5) reticulocyte lysate. Samples were separated with (lane 1) or without (lanes 2–5) prior reduction with DTT.

**Figure 5. Cytosolic TrxR1 activity is required to allow correct disulfide formation in proteins translocated into the ER.**

A Translation of β1-integrin was carried out in the presence of SP cells and the absence or presence of G6P as indicated. Samples were separated with or without prior reduction with DTT as indicated. Auranofin (aur; 20 μM) was either included in the reticulocyte lysate (L) during translation (lane 7) or the SP cells (C) were pre-treated with auranofin (lane 8).

B Translations of β1-integrin carried out in the presence of SP cells (lanes 1 and 2) were subjected to immunoisolation with the conformation-specific antibody 9EG7 (lanes 3 and 4). Translations were carried out in the absence and presence of G6P as indicated and samples separated under non-reducing conditions.

C As in (A), but carmustine (carm; 1 mM) was used instead of auranofin.

mobility of the protein synthesized in the presence of G6P. The untranslated protein seen when SP cells are included in the translation in the presence of G6P migrates at the same position as the NEM-modified protein, further demonstrating the lack of disulfide formation (Fig 5A, lane 6). Taken together, these results indicate that the translocated, glycosylated β1-integrin synthesized in the presence of SP cells formed a different set of disulfides in the presence of G6P than in its absence.

To determine whether native disulfides were formed in either the presence or absence of G6P, we used a conformation-specific antibody (9EG7) which we have shown previously to immunoisolate only β1-integrin that is correctly folded (Tiwari *et al*, 2011). Only

the translation product synthesized in the presence of G6P was immunoisolated by this antibody (Fig 5B) confirming that G6P added to the cytosol was required to ensure the correct disulfides were formed in β1-integrin. These results indicate that non-native disulfides are formed in the translocated, glycosylated product in the absence of G6P and that the NADPH recycled after G6P metabolism is required to resolve these non-native disulfides to allow the protein to fold correctly.

The reconstitution of the ER reductive pathway following G6P addition could be due to the production of NADPH in the cytosol, as shown above, or due to NADPH production within the ER following G6P import and conversion to 6-phosphogluconate by the ER-resident enzyme hexose 6-phosphate dehydrogenase (White *et al*, 2007). The NADPH produced could then be used by any putative GR or TrxR within the ER lumen (Bulleid & van Lith, 2014). To evaluate these two possibilities, first we carried out translation in the presence of SP cells and G6P and inhibited the cytosolic TrxR1 or GR with auranofin or carmustine respectively. When added to the reticulocyte lysate, auranofin blocked the effect of G6P addition (Fig 5A, lanes 6 and 7) whereas carmustine had no effect (Fig 5C, lanes 6 and 7). We then carried out a translation using SP cells that had been pre-treated with auranofin or carmustine. As both these inhibitors are irreversible, any putative ER-localized thioredoxin or glutathione reductase would be inactivated. When these SP cells were added to untreated lysate either in the presence or absence of G6P, no effect of either inhibitor was seen on the ability of G6P to prevent non-native disulfide formation (Fig 5A and C, lanes 6 and 8). These results demonstrate that the cytosolic TrxR1 but not the GR pathway is required to resolve any non-native disulfides formed during the synthesis and folding of β1-integrin following its translocation into the ER lumen.

### The thioredoxin reductase pathway is required for efficient trafficking of proteins secreted from mammalian cells

The results presented above provide compelling evidence that the cytosolic thioredoxin system is required to resolve non-native disulfides within a functionally and morphologically intact ER. To determine whether a similar requirement is evident in intact mammalian cells grown in culture, we made use of well-characterized folding and secretion of the LDLr which has been shown to form non-native disulfides as part of its normal folding pathway (Jansens *et al*, 2002). To determine the role of cytosolic thioredoxin reductase (TrxR1) in the folding and secretion of LDLr, we treated cells with low concentration (5 μM) of the irreversible TrxR inhibitor auranofin (Fig 6). We determined the consequence of TrxR1 inhibition on LDLr folding and trafficking by evaluating the appearance of the O-linked glycosylated form of the protein which has a decreased electrophoretic mobility compared to the ER-localized protein (Oka *et al*, 2013). Trafficking of the LDLr was followed by pulse chase analysis. In the control cells, the appearance of the Golgi form of the protein occurred by 30 min with most of the labeled protein having left the ER by 60 min (Fig 6A). In contrast, in the auranofin-treated cells, no Golgi form was seen after 30 min with the majority of the protein remaining in the ER even after 60 min (Fig 6B). In the presence of auranofin, the ER-localized protein remained as a diffuse band indicative of the formation of non-native disulfides. This result illustrates a

dramatic effect on protein trafficking indicating a lack of correct folding and retention of LDLr in the ER following treatment of cells with a TrxR inhibitor.

Auranofin is generally believed to inhibit TrxR1 by modification of the selenocysteine (Sec) residue within its active site, although other modes of inhibition have also been suggested (Gromer *et al*, 1998; Angelucci *et al*, 2009; Lothrop *et al*, 2009). Interestingly, inhibition of TrxR1 by chemical targeting of its Sec residue can often lead to a gain of function of TrxR1, with an ability to act as a NADPH oxidase thereby generating superoxide or hydrogen peroxide (Anestal *et al*, 2008). An increased generation of hydrogen peroxide could lead to compromised protein folding in the ER. To address this point, we also inhibited thioredoxin reductase with a novel inhibitor (TRi-2) that was recently found to inhibit TrxR1 with similar potency as auranofin and without inhibiting glutathione reductase (Fig EV2) but importantly without the oxidase gain of function (Stafford, W. *et al*, manuscript in preparation). Crucially in contrast to auranofin, treatment of cells with TRi-2 does not lead to an increased production of hydrogen peroxide (Stafford, W. *et al*, manuscript in preparation). When we treated cells with TRi-2 and carried out a pulse chase analysis of LDLr trafficking and folding, again we saw a significant reduction in the trafficking of LDLr, with the ER retained form migrating as a diffuse band (Fig 6C). The effect is not as pronounced as auranofin which could be due to more efficient inhibition of TrxR1 in cells with auranofin or the production of hydrogen peroxide contributing toward the defect in protein trafficking. However, the results do suggest that the effects of TrxR1 inhibition on LDLr trafficking cannot be totally explained by increased hydrogen peroxide production by selenium compromised forms of the enzyme. It should be noted that both auranofin and TRi-2 can trigger cell death upon prolonged exposure, but not within the shorter time frame utilized in our studies (Arner & Holmgren, 2000). The effects seen here on protein trafficking are likely to be direct consequences of TrxR1 inhibition.

Finally, we determined the trafficking and secretion of a protein lacking disulfides, α1-anti-trypsin, in the absence and presence of auranofin to see whether the inhibitor causes a general block in secretion. There was no difference in the rate of appearance of the Golgi localized form or in the rate of secretion following auranofin treatment (Fig 6D and E). Taken together, our results suggest that TrxR1 activity is required for correct folding of proteins entering the ER, in the reconstituted translation/translocation system as well as in intact cells grown in culture.

## Discussion

The results from this work demonstrate that the cytosolic reductive pathway, specifically cytosolic TrxR1, is required for the reduction in non-native disulfides within proteins entering the ER. To investigate this process, we took advantage of the fact that non-native disulfides were formed in proteins with complex disulfide linkages in a rabbit reticulocyte lysate that could be resolved into native disulfides upon the addition of G6P. We demonstrate that added G6P recycles NADPH providing an electron donor for cytosolic TrxR1, which in turn reduces Trx1. A robust TrxR1 pathway is required to prevent the formation of non-native disulfides and, importantly, allow the correct disulfides to form within proteins

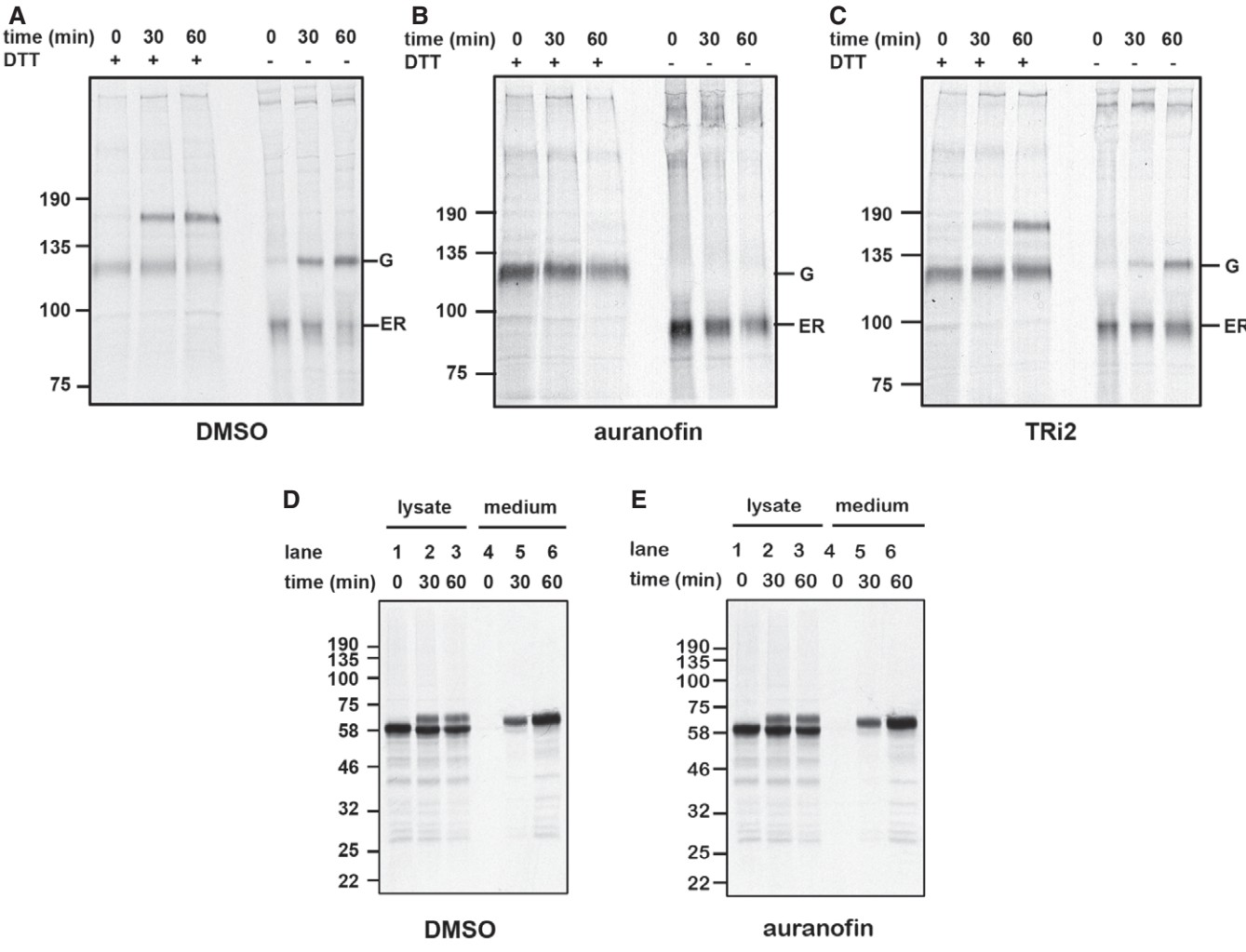

**Figure 6.  Inhibition of TrxR1 compromises LDLr folding and trafficking.**

A–C  HT1080 cells were metabolically radiolabeled with $^{35}$S for 30 min and then chased in the absence of radiolabel for up to 60 min as indicated in the presence of vehicle (DMSO) (A), 5 μM auranofin (B), or 5 μM TRi2 (C). Radiolabeled LDLr was immunoisolated with anti-LDLr and samples separated before or after reduction as indicated. The position of the Golgi (G) and ER forms migrating under non-reducing conditions are as indicated.

D, E  HT1080 cells transfected with V5-tagged α1-antitrypsin were radiolabeled and cells chased in the absence (D) or presence (E) of auranofin as in (A–C) and radiolabeled products immunoisolated with a V5-antibody.

entering the ER. This is the first indication that such a link with TrxR1 exists and evokes comparison between the mechanism of disulfide formation in the ER and the periplasm of Gram-negative bacteria. In bacteria, the membrane protein DsbD shuttles electrons across the periplasmic membrane providing the connection between the two distinct compartments (Kadokura *et al*, 2003). No such membrane protein has been identified in mammalian cells but the logical conclusion from our results is that such a protein exists and will play a key role in driving the reduction in members of the PDI family to facilitate correct protein folding.

The most novel and unique aspect of this work is that we have been able to assay the transition from non-native to native disulfides using an *in vitro* translation translocation system that reproduces the early stages of secretory protein biogenesis. This assay contrasts with other approaches which monitor the thiol/disulfide status of proteins with reversible disulfides such as redox-sensitive GFP (van

Lith *et al*, 2011; Avezov *et al*, 2013) or members of the PDI family (Appenzeller-Herzog *et al*, 2010). The distinction between the assays is apposite; the former assay distinguishes two different types of disulfide whereas the latter distinguishes between reduced or oxidized protein. Establishing conditions that supported the formation of disulfides but prevented the resolution of non-native disulfides allowed us to determine the factors required to catalyze the transition to the native disulfides.

Importantly, our results do not rule out a role for GSH in buffering redox conditions in the ER (Bulleid & Ellgaard, 2011); rather they address the source of the electron donor driving the catalysis of non-native disulfide reduction. A number of studies have demonstrated a critical role of GSH for maintaining ER redox homeostasis. In particular, it is required to re-establish redox conditions following reduction or oxidation of the ER (Molteni *et al*, 2004; Chakravarthi *et al*, 2006; Appenzeller-Herzog *et al*, 2010). In

addition, it is thought to play a crucial role in regulating the activity of Ero1 by facilitating the reduction of PDI (Cuozzo & Kaiser, 1999; Appenzeller-Herzog *et al*, 2010; Kim *et al*, 2012). In contrast to these roles, depletion of GSH has little or no effect on disulfide formation or isomerization of non-native disulfides as the LDLr folds and is trafficked normally in cells depleted of ER GSH (Tsunoda *et al*, 2014), and as shown herein for β1-integrin.

In addition to providing insight into the fidelity of disulfide formation in the ER, we also address the question of how disulfides are prevented from forming within proteins synthesized in the cytosol. The role of GSH in preventing disulfide formation in the yeast cytosol has recently been questioned due to a lack of effect on thiol-redox maintenance following its depletion (Kumar *et al*, 2011). Our results demonstrate that the Trx/TrxR1 system is the primary reductive pathway that prevents disulfides forming in the cytosol with even a low GSSG/GSH ratio unable to prevent disulfide formation. Such a requirement agrees with the lower reduction potential of thioredoxin (−270 mV; Lundstrom & Holmgren, 1993) compared to GSH (−240 mV; Schafer & Buettner, 2001) and may also reflect the ability of Trx1 to bind to oxidized substrates, thereby providing a kinetic as well as a thermodynamic advantage over GSH. Indeed, it has recently been shown that Trx1 preferentially binds to conformationally restricted molecules such as those containing non-native disulfides (Palde & Carroll, 2015). The ability of Trx1 to preferentially reduce disulfides in proteins supports the notion that cytosolic proteins whose function is regulated by disulfide formation are recycled by the Trx1 rather than GSH pathway (Toledano *et al*, 2013).

In summary, our results provide new insight into how redox homeostasis is regulated in the mammalian ER and highlight the presence of a distinct pathway for non-native disulfide bond resolution. The location of the source of reducing power is compartmentally distinct from the ER raising the probability that electron transfer across the ER membrane regulates oxidative protein folding. The identification of NADPH as the electron donor also provides a link between glucose metabolism and protein folding which could influence correct protein folding and cellular stress responses following fluctuations in glucose availability. The challenge now will be to identify the cellular components that connect the activity of TrxR1 in the cytosol with the reduction in the PDI family members such as ERdj5 in the ER lumen.

## Materials and Methods

### Expression constructs, cell lines, and reagents

The plasmid vector encoding V5-tagged α1-antitrypsin, influenza virus HA, bovine preprolactin, and cVIMP-cys were gifted from Lisa Swanton (University of Manchester), Mary Jane Gething (University of Melbourne), Stephen High (University of Manchester), and Lars Ellgaard (University of Copenhagen), respectively. The β1-integrin, hemagglutinin, and the soluble Sec-Cys variant of SelS/VIMP (cVIMP-cys) sequences have been described previously (Gething *et al*, 1980; Tiwari *et al*, 2011; Christensen *et al*, 2012). Auranofin, carmustine, and DHEA were purchased from Sigma-Aldrich; TRi2 synthesis will be described elsewhere (Stafford, W. *et al*, manuscript in preparation). The commercial antibodies used

were anti-His (Proteintech), anti-V5-tag (Invitrogen), and 9EG7 (BD Bioscience). Antibody to LDLr (121) was a gift from Ineke Braakman (Utrecht University; Pena *et al*, 2010). Purified recombinant ChaC1 enzyme was a gift from David Ron (University of Cambridge; Tsunoda *et al*, 2014). The plasmid containing human thioredoxin (hTrx) with a streptavidin binding peptide (SBP) tag was a gift from Tobias Dick (German Cancer Research Centre, Heidelberg). Recombinant Trx protein was purified as described previously (Schwertassek *et al*, 2007).

### Translation reactions

DNA was transcribed and proteins translated essentially as described previously (Jessop *et al*, 2007). Human β1 integrin was transcribed using SP6 polymerase from a pSPUTK vector linearized with EcoRV. A volume of 16.5 μl Flexi rabbit reticulocyte lysate (RRL; Promega) was aliquoted and the following reagents added to a final volume of 25 μl at the indicated final concentrations: KCl at 40 mM, amino acids minus methionine (Promega) at 20 μM and EasyTagTM EXPRESS$^{35}$S Protein Labelling Mix (PerkinElmer), containing both $^{35}$S-L-methionine and $^{35}$S-L-cysteine for radiolabeling of synthesized proteins at 16.3 kBq/μl. Except where stated otherwise, the lysate mixture was also supplemented with G6P at a final concentration of 0.5 mM. RNA transcript was added to initiate translation. All translation reactions were incubated at 30°C in a water bath for 60 min. Reactions were terminated by the addition of the thiol alkylating agent N-ethylmaleimide (NEM; Thermo Scientific) to a final concentration of 50 mM (Braakman *et al*, 1992). For RRL samples containing no SP cells, 2 μl of the lysate sample was added to SDS–PAGE loading buffer. Samples containing SP cells were pulse centrifuged at $12,470 \times g$ for 20 s to pellet the cells. The lysate was then removed and 100 μl of KHM buffer (20 mM HEPES buffer pH 7.2, containing 110 mM KOAc, 2 mM MgOAc) was added. The pulse centrifugation was repeated, and the KHM buffer was removed and SDS–PAGE loading buffer was added. The samples were separated by SDS–PAGE. The gels were then fixed with 10% acetic acid and 10% methanol solution for 20 min, dried, and exposed to either a phosphorimager plate or a BioMax MR film (Kodak); the images captured using a phosphorimager plate were scanned using a FLA-7000 bioimager (Fujifilm).

### Assessing the redox status of thioredoxin

Standard translation reactions were supplemented with SBP (streptavidin binding peptide)-tagged hTrx (3.6 μM; Schwertassek *et al*, 2007) in the presence and absence of G6P (0.5 mM) or 20 μM auranofin and incubated at 30°C for 60 min followed by alkylation with 40 mM NEM. Tagged thioredoxin was then isolated using streptavidin agarose beads and eluted with 3 mM biotin in 2% SDS and boiled for 15 min. Disulfides in isolated hTrx were reduced with 10 mM TCEP and alkylated with 25 mM AMS for 1 h in the dark. Sample was then analyzed by immunoblotting using an anti-His antibody.

### Generation of SP cells and treatment with inhibitors

SP cells were generated from HT1080 cells as described previously (Wilson *et al*, 1995). Approximately $10^5$ cells were added per 25 μl

translation reaction. The enzymes G6PDH, TrxR, and GR were inhibited in the rabbit reticulocyte lysate by addition of DHEA, auranofin or carmustine, respectively, at the indicated concentrations and using a volume of solvent < 4% of total. The translation mixture was incubated for 10 min on ice before translation reactions were initiated by adding mRNA and incubating the mixture at 30°C in a water bath. Where a pre-incubation of inhibitor with SP cells was carried out, cells were isolated free of inhibitor by centrifugation prior to resuspension in the translation reaction.

### Glutathione depletion from reticulocyte lysate, GSH, GR, and Trx assays

Reticulocyte lysate was depleted of GSH by incubation with purified ChaC1 (5 μM) at 20°C for 1 h. Glutathione was assayed essentially as described previously (Tietze, 1969). Glutathione reductase was assayed as described using recombinant human GR (Sigma; Liu & Sturla, 2009). Thioredoxin reductase activity was measured using a Trx1-linked spectrophotometric insulin reduction assay according to manufacturer's instructions (IMCO Corporation). Inhibitors were pre-incubated with Trx and TrxR1 for 15 min before the start of the reaction.

### Metabolic labeling and pulse-chase analysis

HT1080 cells were incubated in medium lacking methionine and cysteine for 30 min and pulse labeled for 30 min with 11 μCi/ml of Express 35S Protein Labelling Mix (Perkin Elmer). The radio-label was removed with PBS washes. The cells were then incubated in complete medium to initiate the chase periods. To monitor the effects of inhibition of TrxR1 on folding and secretion of endogenous LDLR, cells were incubated either with or without 5 μM auranofin or 5 μM Tri2, during starve, label, and chase period. Following the chase period, cells were washed twice with PBS supplemented with 20 mM NEM and lysed in 50 mM Tris–HCl buffer, pH 7.4 containing 1% (v/v) Triton X-100, 150 mM NaCl, 2 mM EDTA, and 20 mM NEM. Postnuclear supernatant was obtained by centrifugation at 4°C, and immunoisolation was performed by pre-clearing the supernatant with protein A Sepharose for 30 min, followed by incubation with the anti-LDLr antibody (121) and protein A Sepharose for 16 h at 4°C. Beads were washed three times in lysis buffer before analysis by SDS–PAGE. Gels were fixed, dried, and radiolabeled protein visualized by autoradiography.

HT1080 cells transfected with wild-type V5-tagged A1AT incubated in the absence or presence of 5 μM auranofin were pulse labeled and immunoisolated with a V5-antibody as described above.

**Expanded View** for this article is available online.

## Acknowledgements

We wish to acknowledge the generosity of all our colleagues for contributing reagents and other members of the Bulleid group for critical reading of the manuscript. This work was funded by the Wellcome Trust grant number 103720, the BBSRC grant ref: BB/L00593 and BB/F016735, and the Swedish Cancer Society and Karolinska Institutet.

## Author contributions

The work described in this manuscript was conceived and supervised by NJB with contribution from GJP. The experimental work was carried out by GJP, OBVO, MvL, ZC, PJR, MAP, and NJB. The data were analyzed by GJP, NJB, and ESJA. The manuscript was written by NJB with editing by all authors.

## Conflict of interest

The authors declare that they have no conflict of interest.

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
