## [Review Process File · The EMBO Journal]

Manuscript EMBO-2016-95336

Cytosolic Thioredoxin Reductase 1 is required for Correct Disulfide Formation in the ER

Greg Poet, Ojore Oka, Marcel van Lith, Zhenbo Cao, Philip Robinson, Marie Anne Pringle, Elias Arnér and Neil Bulleid

Corresponding author: Neil Bulleid, University of Glasgow

Review timeline:

Submission date:	25 July 2016
Editorial Decision:	24 August 2016
Revision received:	28 November 2016
Editorial Decision:	21 December 2016
Revision received:	22 December 2016
Accepted:	22 December 2016

Editor: Andrea Leibfried

Transaction Report:

1st Editorial Decision

24 August 2016

Thank you for submitting your manuscript for consideration by the EMBO Journal. It has now been seen by three referees whose comments are shown below.

As you will see, your work received mixed reports from the referees. Both referee #1 and #3 support publication of your work pending satisfactory revision, while referee #2 thinks that the work is too preliminary and would need more mechanistic insight. All referees offer constructive input on how to improve your work, and I would thus like to invite you to submit a revised version of the manuscript, addressing the comments of all three reviewers.

Importantly, the control requested by referee #1 (point 2) needs to be added for publication here. I don't know whether you have additional data at hand to respond to referee #3's criticism regarding lack of mechanism, thus please let me know in case you want to discuss the revision further with me. I should add that it is EMBO Journal policy to allow only a single round of revision, and acceptance of your manuscript will therefore depend on the completeness of your responses in this revised version.

REFeree REPORTS

Referee #1:

This manuscript investigates a central question in mammalian redox biology, relating to the origin of reducing equivalents needed for disulfide isomerization in the ER. Various possible pathways for disulfide reduction in the ER have been suggested, but the exact molecules/proteins involved have not yet been identified to firmly establish this pathway.

The authors first demonstrate, in large part through the use of chemical inhibitors of the two main cytosolic pathways for disulfide reduction, that the thioredoxin reductase/thioredoxin pathway (rather than the glutathione reductase/GSH/Grx pathway) can prevent disulfide-bond formation in the cytosol, in a process dependent on NADPH. Using semi-permeabilized cells and β 1-integrin as a model protein, the authors then show that correct oxidative folding of this protein in the ER requires addition of glucose-6-phosphate to the lysate (i.e. the cytosol). In a crucial experiment, inhibition of thioredoxin reductase (but not glutathione reductase) in the cytosol indicated that the reducing equivalents provided by the former protein are required for native disulfide formation in the ER. Finally, using the LDL receptor that is known to depend on disulfide isomerization for productive folding, it was shown that transport of the protein from the ER to the Golgi occurred with significantly slower kinetics in the presence of two different thioredoxin reductase inhibitors. These data thus indicated that the LDL receptor did not fold correctly when cells were treated with thioredoxin reductase inhibitors.

Conceptually, the original data provided here change how we think about disulfide-bond formation in the ER. Although all proteins involved in the relay of reducing equivalents from the cytosol to the ER are not yet known, this is a major advance in the field. The results are clear and the conclusions are generally supported by the data. Still, I have the following concerns to be addressed:

1. Page 11, Fig. 5D/E:

In this crucial experiment, SP cells are treated with the inhibitors auranofin and carmustine. Thus, the outcome of the result relies completely on these inhibitors being able to cross the ER membrane. Can the authors provide evidence that this occurs in their experimental set-up? This point also relates to the results presented in Fig. 5D, lane 8. Here, the authors conclude that there is no effect of pretreating SP cells with auranofin. Still, the signal in lane 8 is significantly lower than the signal in lane 6. Could this potentially represent a certain effect of auranofin in the ER? If so, a higher concentration of auranofin might have a larger effect - has this been tested?

In addition, the interpretation of the experiments in Fig. 5 also relies on NADPH not crossing the ER membrane in this experimental set-up - is this the case?

2. Page 12-13, Fig. 6:

This experiment is clear and convincing, and shows that ER-to-Golgi transport of the LDLR is impaired in the presence of thioredoxin reductase inhibitors. Potentially though, this could be a general impairment of ER-to-Golgi transport caused by inhibitor treatment. It would therefore strengthen the conclusion to include a control experiment that shows no impairment of ER-to-Golgi transport for a protein devoid of disulfide bonds.

3. Page 14, Discussion:

The authors state at the beginning of the Discussion that cytosolic TrxR1 is required for the reduction of non-native disulfides within proteins entering the ER. This point is shown for β 1-integrin, but not directly for LDLR. It would therefore strengthen the conclusion to show directly in a pulse-chase folding assay (rather than the more indirect transport assay) for LDLR that the isomerization required for productive folding of this protein is impaired when treating cells with thioredoxin reductase inhibitors.

Minor points:

p. 2: At the end of the Abstract the authors are almost being too modest. I think it would be justified to more clearly state why the findings are conceptually new (the non-expert might not completely understand why it is interesting that electrons are relayed into the ER as they are into the periplasm of certain prokaryotes, and it also makes the current data sound more "ordinary").

Three different abbreviations are used for thioredoxin reductase: TrxR, TrxR1 and TXNRD1.

Likewise, thioredoxin is abbreviated in two different ways: Trx and Trx1.

The same paper is referenced twice (Bulleid NJ, Ellgaard L (2011a) and Bulleid NJ, Ellgaard L (2011b)).

p. 3: Concerning the sentence: "In addition, disulfides in proteins that are targeted for degradation need to be reduced prior to their dislocation from the ER to the cytosol (Ushioda, Hoseki et al., 2008), the word "need" could be debated, but certainly examples exist that this seems to be the case. Further references could be added, such as He K, et al. *MBC*, 2015; 26(19):3413-23.

p. 5: A last possibility to get rid of ER-generated GSSG would be to transport it back to the cytosol (or export it from the cell).

p. 8: The information that purified tagged Trx was added to the lysate would benefit from further explanation, i.e. which tag was used and for which purpose. This point also relates to the Fig. 4 legend.

p. 8: It would be helpful for the reader if the reason/background for performing the two sets of experiments starting with the sentences: "We also monitored the change in GSSG/GSH ratio following G6P addition" and "Finally, to determine the consequence of the removal of GSH from the...." was mentioned before writing about the results.

p. 9: It is a bit unclear why the authors cite the Reichheld paper the way they do, since their own data show no indication of crosstalk between the two pathways..?

p. 15: Concerning the sentence: "In addition, it is thought to play a crucial role in regulating the activity of Ero1 by facilitating the reduction of PDI (Bulleid & Ellgaard, 2011b, Cuozzo & Kaiser, 1999)", it would be appropriate to also cite Kim et al. *J Cell Biol.* 2012 Mar 19;196(6):713-25. Moreover, it would be more suitable to cite experimental work for the mammalian system, rather than the review currently cited.

p. 29, Fig. 2: It would be illustrative to also include in panels A and B (on the left) how G6PDH generates NADPH using G6P.

p. 30, Fig. 3: It is unclear why there is seemingly a quite large migration difference between the material in Fig. 3A, lane 1 (just below 100 kDa) and Fig. 3B, lanes 1 and 2 (just below 80 kDa). Likewise, it is unclear why oxidized β 1-integrin migrates differently in Fig. 3A, lane 3 and Fig. 4C, lane 2.

p. 32, Fig. 5: It would be helpful if marker bands were included in all panels for comparison with other figures. Also, it would be good to indicate the migration of the upper bands in panel B, lanes 2 and 4 to illustrate the difference in migration.

p. 35, Fig. S1: It would be useful to add indicator lines at the side of each panel to mark the mobility of oxidized and reduced HA.

In this experiment it is unclear why HA does seemingly only completely oxidise in the presence of high concentrations of DHEA, but not with high concentrations of auranofin, where the reduced form presumably never disappears. Can this be rationalized?

Suggestions for language revisions:

p. 2: Insert "in the ER" after "Here we use a novel assay to demonstrate that the reduction of non-native disulfides".

p. 2: Insert "the disulfide isomerase" in front of "DsbC".

p. 3: Remove the "s" at the end of "dithiol/disulfides".

p. 4: Insert "in turn" after "...allowing de novo disulfide formation in the disulfide exchange protein DsbA which".

p. 4: Insert "in substrate proteins" after "the disulfide exchange protein DsbC that finally catalyses disulfide reduction".

p. 5: Insert "ER" after "To provide insight into the reductive pathway involved in ensuring that correct disulfides are formed within".

Referee #2:

The source of reductive force in the oxidative condition of the ER is an important and interesting issue to be addressed. Poet et al. describe here that NADPH is the ultimate electron donor for the correct folding of the nascent proteins synthesized in the ER. To show it, the authors adopted an in vitro translation system, which lacks DTT, and SP cells in combination with the use of several inhibitors for G6PDH, TrxR1 and GR. The authors showed that addition of G6P inhibited the disulfide formation in the folding assay using c-VIMP, HA, pre-PL and b1-integrin, and the inhibitors of G6PDH and TrxR1 but not GR canceled the effect of G6P, which suggested NADPH is the source of reduction of substrates via TrxR1 and Trx pathway. They also showed that GSH is not involved in the G6P-mediated reduction of disulfides by using ChaC1, a GSH-degrading enzyme. While the results showed here are clear, the experiments presented here sound too preliminary for publication in EMBO J, and several experiments did not focus on and support the conclusion that the authors wanted to derive from the experiments. In addition, major parts of this manuscript had been reported by other groups including D. Ron (GSH is not directly involved in the reduction) and I.J. Benjamin (G6PDH).

1. Fig. 1 shows only the validity of the experimental system because the ER luminal proteins were translated without addition of microsomes, and did not provide any useful information on the conclusion. Fig. 1 only shows that NADPH can reduce the ER resident proteins as well as cytosolic proteins. This should be shown as supplementary data.
2. Fig. 2 is not necessary, at least not in the result section.
3. Only in Fig. 6, the LDL receptor was used, why? Authors possess the useful antibody to discriminate folded and unfolded LDL receptors, which provide clear results and conclusions. Why was this useful method not adopted throughout the experiments?
4. In the previous paper by this group, the involvement of ERdj5 was shown in the folding of LDL receptors. If so, the electron transfer from NADPH to ERdj5 and then to LDL receptors can be more clearly shown. In general, the paper described only phenomenological observation such as folding of the substrates and reduced or oxidized forms of the substrates, but not the mechanical aspect.
5. The most interesting issue of this manuscript might be the mechanism how the reductive force in the cytosol can be transferred to the ER lumen, which was not addressed in this manuscript.

Referee #3:

It is generally appreciated that the endoplasmic reticulum (ER) must retain a redox balance that supports disulfide isomerization and re-reduction as well as net disulfide bond formation in proteins folding in that compartment. It was long assumed that the entry of reduced glutathione into the ER, plus the import of reduced cysteines in nascent proteins, provided the reducing power needed to correct non-native disulfide formation. However, it was recently shown that inducing degradation of glutathione in the ER had no effect on the folding of proteins known to require disulfide rearrangements for maturation. This finding left open the question of what other source of reducing power may be available to the ER.

The manuscript under consideration takes a significant step in addressing that question. Starting with in vitro translation (TL) experiments, Poet et al. demonstrate that fueling the pentose phosphate pathway by addition of G6P to the in vitro TL mix helps resolve non-native disulfides in the translated proteins. The pentose phosphate pathway supplies NADPH, which is used by thioredoxin reductase (TrxR1) to reduce thioredoxin (Trx). Experiments in semipermeabilized and intact cells are then performed to show that interfering with TrxR1 activity also interferes with native disulfide bond formation in proteins co-translationally translocated (presumably) into the ER, despite the fact

that TrxR1 and Trx are cytosolic. It was previously noticed that treatment with auranofin, the main compound used to inhibit TrxR1 in this study, induced the ER stress response (Fiskus et al., Cancer Research, 2014), but this finding was assumed to be due to an increase in ROS upon auranofin treatment (Gandin et al., Biochem. Pharmacol. 2010). In an attempt to confirm that the primary TrxR1 activity (reduction of Trx) is the cause of the phenomena occurring in the ER lumen, Poet et al. use a new inhibitor of TrxR1 that they report (citing a manuscript in preparation) does not cause TrxR1 to produce ROS.

Ideally, one would like to know HOW TrxR1 activity enables reducing equivalents to enter the ER and be utilized. However, it can fairly be said that the full revelation of this mechanism would be beyond the scope of the current paper.

The experiments and results that are within the scope are for the most part presented logically and convincingly. One possible exception is Figure 6C, in which the effect of Tri-C is quite moderate, which seems to contradict its potent effect on thioredoxin reductase activity presented in Fig. 2S. Is there a possible role for ROS as well as Trx in the effects seen with auranofin? Or some other explanation?

There are a few mistakes, insertions, or omissions in the text:

The first word in the abstract seems unnecessary and misleading. Efficiency doesn't seem to be the real issue here.

In the discussion, it is stated, "... G6P recycles NADPH providing an electron acceptor for cytosolic TrxR1..." Shouldn't that be electron donor?

What is KHM buffer?

1st Revision - authors' response

28 November 2016

We detail below our point-by-point responses to the reviewers' comments.

Referee #1:

1. Page 11, Fig. 5D/E:

The referee has two concerns regarding this particular figure. The first is that inhibitors may not pass across the ER membrane to inhibit any potential reductases and the second is that NADPH may pass across the ER membrane. Both auranofin and carmustine have been used extensively as reagents to inhibit either thioredoxin reductase or glutathione reductase in cells grown in culture. They can both pass across the plasma membrane so it is highly unlikely that they cannot pass across the ER membrane. The referee is correct to point out that there is less product in Fig. 5 lane 8 than in lane 6. We are using an auranofin concentration (20 μ M) known to inhibit thioredoxin reductase completely (Fig. 3B) and the additional incubation of the SP-cells does have a variable effect on translocation efficiency. However, we are looking at a qualitative result in that there is no inhibition of the effect of G6P which is clearly seen when auranofin is added to the reticulocyte lysate.

The permeability of the ER membrane to the water-soluble electron carriers NADP or NADPH has been addressed previously (Piccirella et al, JBC, 281, 4671-4677). The ER membrane was shown to be essentially non-permeant to either NADP or NADPH.

2. Page 12-13, Fig 6:

We have now carried out a secretion time course with a non-disulfide bonded protein, alpha-1-antitrypsin. As can be seen (Fig. 6 D, E) there is no effect of auranofin on transport from the ER or on protein secretion. We can conclude that auranofin does not cause a general impairment of protein trafficking.

3. Page 14, Discussion:

The first line in the discussion states that we conclude from all our results that cytosolic thioredoxin reductase is required for the reduction of non-native disulfides within proteins entering the ER. The

referee correctly points out that in our original submission this refers to our results with β 1-integrin. We have also shown data for the persistence of non-native disulfides in the endogenous LDLr receptor expressed in the presence of auranofin in this submission.

Minor points

P2. We have adjusted the abstract to more clearly state why the findings are conceptually new. We have adjusted the abbreviations where appropriate. Trx is used for thioredoxin (for example thioredoxin domains), Trx1 is used for a specific thioredoxin isoform found in the mammalian cytosol and TrxR1 is used for thioredoxin but is also abbreviated to TXNRD1 in the literature so the two abbreviations are included the first time used but only TrxR1 is used subsequently. The duplication of the citation has been resolved.

P3. This sentence has been changed and further reference added.

P5. This additional possibility has been included.

P8. We have added additional text to include the reason for these experiments prior to describing the results.

P9. The reference is to indicate that previous results do indicate a crosstalk between the pathways. The sentence has been modified accordingly.

P15. These citations have been modified

P29. We have modified Figure 2 as suggested.

P30. The migration differences were due to different markers used. We have repeated the experiment in Fig. 3B with the same markers as 3A.

P32. We have included markers for all panels. We have rearranged the panels in Figure 5. You can now note a clear difference in mobility between Fig. 5 lanes 2 and 6.

P35, Fig S1. We have included the indicator lines to depict the reduced and oxidised HA. We have also replaced the panel in B with a repeat that more clearly demonstrates the complete oxidation of HA in the presence of auranofin.

Referee #2

The referee indicates that the results are preliminary and that the results have been reported before. We contest this view as there is actually no report of the cytosolic pathway being involved in disulfide bond reduction in the ER. The references cited by the referee are looking at different aspects of disulfide formation indicating that GSH is not involved in ensuring correct disulfides in the ER but do not show (as we have) that the thioredoxin pathway is required. These results are novel and while we do not have the complete picture, the impact is high.

Specific points:

1. Figure 1 importantly establishes the fact that disulfide formation in the cytosol can be prevented by the addition of G6P. This is a crucial aspect of the paper and warrants its inclusion in the main body of text.
2. Figure 2 is positioned at the point in the paper that we first discuss the various pathways for reduction in the cytosol. It is good to illustrate the pathways so that readers that are less familiar with the detailed pathways can be orientated regarding our logic flow in the text. This seems to us to be the most appropriate place to include it.
3. We do not have any conformation-specific antibodies to LDLr, they are not commercially available as are the antibodies to β 1-integrin.
4. We would be very keen to now dissect the detailed mechanistic aspects of electron transfer across the ER membrane. This is a focus for our studies now. The current manuscript is a conceptual advance in our thinking of how disulfides are reduced in the ER. The detailed mechanism is beyond the scope of the current manuscript.
5. This seems to be the same comment as the one above.

Referee #3

As mentioned above we are very keen to know how TrxR1 activity enables reducing equivalents to enter the ER and be utilized. The referee agrees that the full revelation of the mechanism is beyond the scope of the current paper. The other comments by the referee are addressed below.

Since we submitted the paper we have carried out several experiments to address the seemingly modest effect of TRi2 on LDLr folding and secretion. We find that the effects can be best demonstrated when looking at endogenously expressed protein rather than by overexpression. We have now replaced the overexpression experiment with a new figure clearly demonstrating an effect of Tri2 on protein folding and secretion. There may well be an additive effect with auranofin producing hydrogen peroxide and we mention this in the text.

We have modified the abstract to take on board the referee's suggestion regarding the first sentence. We have corrected the discussion where it mentions NADPH as an electron acceptor. We include a full description of KHM buffer.

2nd Editorial Decision

21 December 2016

Thank you for submitting your revised manuscript for our consideration. Your manuscript has now been seen once more by two of the original referees (see comments below), and I am happy to inform you that they are both broadly in favor of publication, pending satisfactory minor revision.

I would therefore like to ask you to address the remaining suggestions and to provide a final version of your manuscript.

REFEREE REPORTS

Referee #1:

The current version of this manuscript has addressed all of my original concerns in a satisfactory manner.

Minor points:

- p. 13: in the sentence "..., with the ER retained from...", from should be replaced by form.
- p. 12-13: The word "secretion" in connection with the LDLr is unfortunate, as the protein is not secreted.
- Figures: The authors should consider labeling figure panels more consistently throughout to indicated oxidised, reduced, Golgi-localised, secreted, glycosylated and untranslocated forms of the various proteins.

Referee #3:

The authors addressed many of the points of the referees. There are still a few textual problems, however.

Surely the authors do not mean what they write in the abstract: "Hence, we have shown for the first time that mammalian cells have a pathway for transferring protein disulfides from the cytosol to the ER..."

Also, this reviewer failed to make this point in the first version of the manuscript, but the statement "we can re-establish the reducing environment in the cytosolic extract without the addition of a reducing agent" is misleading. G6P is the source of the reducing equivalents to convert NADP+ to

NADPH and thus functions, via G6PDH, as a reducing agent.

p. 13 "the non-disulfide containing protein α 1-anti-trypsin" can be changed to "a protein lacking disulfides, α 1-anti-trypsin".

2nd Revision - authors' response

22 December 2016

I am pleased to say that we have made the minor corrections requested by the referees and addressed the editorial points.

3rd Editorial Decision

22 December 2016

Thank you for submitting the final version of your manuscript to The EMBO Journal. I appreciate the introduced changes, and I am happy to accept your manuscript for publication.

Corresponding Author Name: Neil Bulleid

Journal Submitted to: The EMBO Journal

Manuscript Number: EMBOJ-2016-95336R